# Vasa Previa and the Role of Fetal Fibronectin and Cervical Length Surveillance: A Review

**DOI:** 10.3390/diagnostics14101016

**Published:** 2024-05-15

**Authors:** Antonia F. Oladipo, Kaitlyn Voity, Kimberly Murphy, Manuel Alvarez, Jesus Alvarez-Perez

**Affiliations:** 1Hackensack Meridian School of Medicine, Hackensack Meridian Health Network, Nutley, NJ 07110, USA; antonia.oladipo@hmhn.org (A.F.O.);; 2Department of Obstetrics and Gynecology, Hackensack Meridian Health Network, Hackensack University Medical Center, Hackensack, NJ 07110, USA

**Keywords:** vasa previa, fetal fibronectin, cervical length, prenatal diagnosis, preterm birth

## Abstract

Vasa previa is a pregnancy complication that occurs when unprotected fetal blood vessels traverse the cervical os, placing the fetus at high risk of exsanguination and fetal death. These fetal vessels may be compromised by fetal movement and compression, leading to poor oxygen distribution and asphyxiation. Diagnostic tools for vasa previa management and preterm labor (PTL) include transvaginal ultrasound, cervical length (CL) surveillance and use of fetal fibronectin (FFN) testing. These tools can prove to be quite useful as they allow for lead time in the prediction of PTL and spontaneous rupture of membranes which can result in devastating outcomes for pregnancies affected by vasa previa. We conducted a literature review on vasa previa management and the usefulness of FFN and CL surveillance in predicting PTL and found 36 related papers. Although there is limited research available to show the impact of FFN and CL surveillance in the management of vasa previa, there is sufficient evidence to support FFN and CL surveillance in predicting the onset of PTL, which can have devastating consequences for the pregnancies affected. It can be extrapolated that these tools, by helping to determine pregnancies at risk for PTL, could improve management and outcomes in patients with vasa previa. Future studies investigating the management of vasa previa with FFN and CL surveillance to reduce the burden of PTL and its associated comorbidities are warranted.

## 1. Introduction

Vasa previa is a pregnancy complication in which unprotected fetal blood vessels traverse the amniotic membranes overlying the internal cervical os, placing the fetus at high risk for exsanguination and fetal death. Risk factors for vasa previa development include abnormal placentation, in vitro fertilization (IVF), multi-fetal gestations, placenta previa or a low-lying placenta and velamentous umbilical cord insertion. Many of these risk factors increase the risk of vasa previa by way of placenta previa or abnormal umbilical cord insertion into the placenta. A low-lying placenta is described as the placental edge being within 0 to 20 mm away from the internal os. A placenta previa occurs when any parts of the placenta overlie the internal cervical os. During the course of a pregnancy, a placenta previa can resolve. Resolution occurs more frequently in the 2nd and early 3rd trimesters. If the placenta remains close to or covering the internal cervical os, near time to delivery, then the likelihood of resolution is low. Abnormal fetal vessels may also remain close to or overlying the internal cervical os [1], describing vasa previa. The likelihood of resolution of vasa previa or a low-lying fetal vessel decreases closer to delivery. Early detection facilitates monitoring and intervention strategies, potentially averting adverse outcomes associated with these conditions. Both IVF and multi-fetal gestation increase the probability of abnormal placentation and placenta previa, thus increasing the risk of developing vasa previa [2]. In pregnancies with multi-fetal gestation, there is an additional increased probability of developing velamentous cord insertion. Velamentous cord insertion occurs when the umbilical cord inserts into any part of the placenta, other than the placental central mass [3]. With this type of cord insertion, there is a higher risk of unprotected vessels with minimal Wharton’s jelly traveling along the placental fetal surface from the point of the placental insertion, increasing the probability of vasa previa development [4].

The major events that impact fetal outcomes in pregnancies with vasa previa are preterm labor and/or premature rupture of the membranes (PROM). These events can lead to fetal demise due to fetal vessel compression and/or vessel rupture [5]. Consequently, timely recognition and management of these events are imperative in mitigating adverse fetal outcomes associated with vasa previa. Early identification of PTL and PROM allows for proactive interventions aimed at optimizing maternal and fetal well-being, thereby reducing the potential for fetal vessel compromise and consequent adverse outcomes.

Current diagnostic tools for PTL include fetal fibronectin (FFN) and a transvaginal cervical length measurement. A cervical length measuring <25 mm places the pregnancy at an increased risk of preterm labor [6]. Cervical length measurement has a positive predictive value (PPV) of 12.5% in asymptomatic women with no past history of preterm labor and 33.5% if preterm birth occurred in the past. The negative predictive value (NPV) is reported to be 86–100% [7]. An FFN test detects the breakdown of proteins that aid in maintaining the attachment of the amniotic sac to the uterus. FFN tests have been shown to have a PPV in symptomatic patients between 43 and 79%. The NPV of symptomatic patients has been reported as 93% for delivery before 34 weeks [8]. In low-risk, asymptomatic patients, the NPV is 95–97% and the PPV is 13–36% [8]. Thus, a negative FFN is more predictive that the likelihood of PTL within the next 7–14 days is low.

## 2. Materials and Methods

A comprehensive literature review was undertaken using the PubMed database, encompassing all time periods. The search strategy employed the following key terms: “vasa previa”, “fetal fibronectin”, “cervical length”, “preterm birth” and “vasa previa diagnostics”. The Boolean operator “AND” was utilized to enhance search precision and acquire pertinent findings. After the initial search, articles not composed in English, those lacking relevance to the subject matter and any duplicates were systematically excluded. Ultimately, a total of 36 papers pertinent to the research theme were identified. Through a meticulous review process, a comprehensive understanding of the existing literature on vasa previa, fetal fibronectin, cervical length, preterm birth and related diagnostic methodologies was established.

## 3. Vasa Previa and Management

### 3.1. Vasa Previa: Definition and Diagnosis

In vasa previa, the fetal blood vessels traverse through the amniotic membrane overlying the internal cervical os. Type 1 vasa previa occurs in the setting of a velamentous cord insertion, when the fetal vessels insert between the amnion and the chorion away from the placenta instead of directly into the chorionic plate, circumventing direct placental insertion [9]. On the other hand, Type 2 vasa previa develops when an extra placental lobe inserts apart from the main placental body. The vessels connecting the bilobed placenta may cross the cervix unprotected [9]. Type 3 vasa previa is when it cannot be classified into the two other types and can be described as when vessels run on the margin of the placenta without velamentous insertion or evidence of a bilobed placenta [10]. Resolution of vasa previa or low-lying fetal vessels can occur in the second and early third trimesters [11]. Classification of vasa previa into distinct types delineates anatomical variations influencing clinical management and prognosis. Timely diagnosis, particularly in relation to gestational age, informs the potential for resolution and guides therapeutic interventions to optimize maternal and fetal outcomes.

There can be a resolution of vasa previa or low-lying fetal vessels closer to the time of delivery depending on the gestational age at which these conditions are diagnosed [11]. Classification of vasa previa into distinct types delineates anatomical variations influencing clinical management and prognosis. Timely diagnosis, particularly in relation to gestational age, informs the potential for resolution and guides therapeutic interventions to optimize maternal and fetal outcomes.

Early diagnosis is critical due to the potential catastrophic complications. A prenatal diagnosis involves a transvaginal ultrasound with color doppler around 18–22 weeks [12]. Fetal vessels can be visualized and characterized in relation to the cervical os, as seen in Figure 1. If the patient has risk factors for vasa previa, such as an IVF pregnancy or a low-lying placenta that has resolved, another transvaginal ultrasound should be performed after 30 weeks for reevaluation [12]. In rare situations, the fetal vessels can be felt upon digital examination [13]. If an antenatal diagnosis of vasa previa is not made prior to labor and the rupture of membranes is followed by bleeding and a non-reassuring fetal heart tracing, vasa previa should be on the differential diagnosis. Outcomes in cases of vasa previa diagnosed during the antenatal period are typically favorable [14].

### 3.2. Vasa Previa: Risks and Management

The major complications that arise from vasa previa are caused by pressure or a disruption to the fetal blood vessels traveling unprotected over the cervix. When there is spontaneous membrane rupture, there is increased pressure put on the vessels, which can result in vessel rupture [1]. Fetal exsanguination from vessel compression and rupture can result in severe hypotension which can be fatal within minutes [5]. Fetal movement can also compress the vessels, resulting in asphyxiation [15]. It has also been noted that pressure placed on unprotected fetal vessels can result in decreased fetal growth [15].

Vasa previa is associated with iatrogenic preterm delivery [13]. Preterm delivery is defined as birth before 37 weeks’ gestation [16]. Iatrogenic preterm delivery refers to a treatment plan where the patient has a planned induction or cesarean delivery prior to 37 weeks. In confirmed cases of vasa previa, it is suggested to have a cesarean delivery between 34 weeks 0 days and 37 weeks 0 days’ gestation [16]. The goal is to deliver the fetus before spontaneous membrane rupture or spontaneous labor onset. However, there is no set clinical protocol, but there are suggested guidelines for the optimal timing of delivery for patients with vasa previa [16]. Part of this is due to balancing the neonatal prematurity risks and the fetal risk associated with vessel rupture. FFN testing in patients with vasa previa may help in better assessing the timing of iatrogenic preterm delivery.

Management of vasa previa before and during delivery is vital to fetal survival [17]. Due to the variations in the presentation of vasa previa, there are no concrete guidelines on the type and amount of surveillance needed, but a personalized care plan is strongly suggested [17]. It has been suggested that diagnosis should be confirmed throughout the pregnancy, and if at 34 weeks the diagnosis remains, then the patient can be admitted to the hospital for close monitoring [12]. After admission, it is recommended to perform at minimum daily non-stress tests until delivery. When the patient has confirmed vasa previa but no other risk factors or signs of labor, outpatient observation can be considered; however, it has been noted that patients who are monitored inpatient have a decreased risk of an emergent cesarean delivery [18]. One can also consider using cervical length measurements to allow for outpatient management and decreased risk of emergent cesarean delivery in vasa previa management [19]. Due to the increased risk of preterm delivery, administration of antenatal corticosteroids in the preterm period should be considered for the benefits to fetal lung maturity [12]. An important consideration when performing a cesarean delivery for vasa previa is to not sharply cut the membranes but use blunt entry at the level of the membranes. By using this technique, the fetal vessels can be identified and avoided until the fetus is delivered [20]. The management of vasa previa at delivery involves the meticulous planning, timing and execution of cesarean delivery to minimize the risk of fetal vessel rupture and optimize maternal and neonatal outcomes. Close collaboration among obstetric, neonatal and anesthesia teams is crucial to ensure the safe delivery of the baby and the well-being of the mother.

## 4. The Use of Fetal Fibronectin in Predicting Labor Outcomes

### 4.1. What Is FFN and Why Is It Important?

FFN is a glycoprotein produced by amniocytes between the maternal and fetal membranes that allows for adhesion [21]. FFN acts as a glue to keep the fetal membranes connected within the uterus. During pregnancy, FFN is produced by the amniotic epithelium and fetal membranes, particularly the chorion. It maintains the structural integrity of the fetal membranes and their attachment to the uterine lining. This adhesive property is crucial for maintaining the pregnancy and preventing premature labor [22]. In an uncomplicated pregnancy, it is expected to see little to no FFN in cervical secretions [21]. FFN is present in the cervicovaginal secretions of pregnant women, particularly during the first and second trimesters. However, its presence in the cervicovaginal secretion between weeks 22 and 34 of gestation is associated with an increased risk of preterm birth. The detection of fetal fibronectin in cervicovaginal secretion during this period is used clinically as a predictive marker for preterm labor [21]. A qualitative FFN test returns either a positive or negative result, which is based on the concentration of FFN noted in the specimen compared to the established threshold. Quantitative FFN allows for a focus on the total FFN concentration instead of measuring if a threshold is met and is reported in a value of ng/mL [23]. Testing for FFN in cervical secretions can provide insight on the maternal and fetal membrane connection stability.

### 4.2. Role of FFN in the Prevention of Preterm Birth

FFN can be clinically useful in predicting and managing preterm labor, as an increase in FFN levels can aid in the prediction of labor onset. An elevated FFN in cervical secretions shows that the connection between the fetus and maternal membranes is dissociating. This dissociation is a risk factor for spontaneous membrane rupture and early delivery [21].

The procedure for testing FFN levels involves a swab inserted into the vaginal canal to collect cervical secretions. The collected specimen is sent to the laboratory for analysis and results can be either positive (detection of FFN) or negative (no detection of FFN) [24]. If the test results return positive, it can be estimated that labor may occur within 7–14 days, but the complete clinical picture should be taken into account as the PPV of the test is 13–36% [24]. The NPV of a negative FFN is 99% and thus, less than 1% of women will deliver within 7–14 days after a negative test. However, it is crucial to note that a false negative is possible, and a negative FFN does not completely rule out preterm labor [25] Fetal fibronectin testing is reported to be highly sensitive (98–100%) with a high NPV (95–97%), while its specificity (64%) and PPV (13–36%) are low [8]. Ultimately, patients are also assessed clinically on symptoms such as contractions, leaking, cramping or pressure, and the FFN test can be used in conjunction with the physical exam for a more thorough assessment. The recommended gestational age to perform FFN testing is between 25 and 34 weeks [26]. FFN testing poses little risk to a pregnancy; however, it is important to note the increased emotions it may provoke for patients surrounding its results, as well as the emotions that come with a false-positive or -negative test result. This may include increased patient stress and anxiety while awaiting the results [27]. Patients may be subject to preterm delivery if a false-positive FFN is interpreted inaccurately. It is important to note that a positive FFN can give insight into the clinical picture but there is not a strong predictive value. Contraindications for FFN testing include anything that can lead to disruption of the vaginal and cervical interface, including placental abruption, premature rupture of the membrane, placenta previa, and moderate to heavy vaginal bleeding. It has also been thought that the use of FFN in managing preterm labor can also help to reduce admissions to the hospital where patients would potentially receive steroids and stay in the hospital until delivery [28].

## 5. Role of Cervical Length Surveillance and the Management of Preterm Delivery

### 5.1. What Is Cervical Length Surveillance and Why Is It Important?

The cervix plays a crucial role in maintaining the integrity of the uterus and supporting a healthy pregnancy. The term “cervical length” refers to the distance between the internal and external cervical os. The technique for obtaining this measurement involves a transvaginal ultrasound with an empty bladder to maintain measurement accuracy [29]. In pregnancy, it is noted that the cervical length decreases as labor approaches [30]. During pregnancy, the cervix undergoes changes in its length and consistency to prepare for childbirth. In the early stages of pregnancy, the cervix is typically long and closed to help support the growing fetus and maintain the pregnancy. As the pregnancy progresses and approaches term, the cervix gradually shortens and softens in preparation for labor and delivery [31]. A change in cervical length or a decreased cervical length around 18 weeks can be a sign or risk of preterm labor [29,30]. The current definition of a short cervical length is less than 25 mm measured at 23 weeks [32]. Monitoring cervical length is important because changes in its length can indicate a heightened risk of preterm birth. By tracking cervical length over time, healthcare providers can identify women at risk of preterm birth early in their pregnancies, allowing for timely interventions such as cervical cerclage or administration of progesterone to help prevent preterm labor.

### 5.2. Role of Cervical Length Surveillance in the Prevention of Preterm Birth

It has been shown that a change in cervical length over time can be closely indicative of preterm delivery [32]. Cervical length surveillance plays a crucial role in identifying women at risk of preterm birth and implementing preventive measures to optimize maternal and neonatal outcomes. Cervical length tends to remain relatively stable during the initial two trimesters of pregnancy. However, relying solely on the natural progression of cervical length change may not effectively identify women at heightened risk of spontaneous preterm birth. This is due to potential variations in the patterns, rates and onsets of cervical length shortening among individuals. Therefore, repeated assessment of cervical length could prove valuable in identifying patients with an elevated risk of spontaneous preterm birth [33]. A change in cervical length can be clinically useful in managing pregnancies that labor poses high risks for, like vasa previa. In vasa previa, the optimal management is to predict and prevent spontaneous membrane rupture and PTL before it happens. Transvaginal ultrasound for cervical length measurement carries minimal risk to the pregnancy; nevertheless, it is crucial to acknowledge the potential emotional impact on patients during the ultrasound and awaiting the results. This may manifest as heightened stress and anxiety, reflecting the significance patients attach to the outcome [34]. Cervical length measurements may aid in the management of patients with vasa previa and help to prolong gestation and optimize timing of delivery [32]. By incorporating cervical length measurements into routine prenatal care, healthcare providers can take proactive steps to mitigate the risk of preterm birth and improve maternal and neonatal outcomes.

## 6. Optimizing Vasa Previa Management with FFN and CL Surveillance

Management of vasa previa is complex due to the high risk of exsanguination of the fetus. However, when considering delivery prior to 37 weeks’ gestation, prematurity complications of the neonate are an important factor in the decision making. Antenatal vasa previa management involves a comprehensive approach in optimizing maternal and fetal benefits in order to prevent early and unpredictable delivery in these patients. Overall, the main goal in management is to prevent fetal vessel rupture, and therefore, these patients are delivered early via cesarean delivery [12]. A potential way to enhance the management of vasa previa is to predict the risk of PTL at different intervals during the pregnancy, specifically through the use of FFN and serial CL surveillance every 1–2 weeks. When used together, these tests increase the predictive value compared to the individual test values alone [35]. A negative FFN and a normal cervical length can, in theory, safely extend the expectant management period as an outpatient until 36–37 weeks, which is not only reassuring to the patient and physician but may also save healthcare costs [3]. Integrating FFN and CL surveillance into vasa previa management protocols enables proactive risk assessment, early intervention and personalized care strategies, ultimately optimizing maternal and fetal outcomes in cases of this rare but serious obstetric complication.

## 7. Discussion

Navigating the complexities of vasa previa management poses a significant challenge, primarily due to the looming threat of fetal exsanguination. However, the decision-making process becomes further complicated when considering delivery before 37 weeks’ gestation, with prematurity-related complications for the neonate becoming a pivotal consideration. Antenatal management of vasa previa demands a holistic approach aimed at maximizing benefits for both mother and fetus, with a key focus on averting early and unpredictable deliveries. Paramount among management goals is the prevention of fetal vessel rupture, often necessitating early cesarean delivery. An emerging avenue to augment vasa previa management involves the proactive prediction of preterm labor (PTL) risk at various stages of pregnancy. This can be achieved through the judicious use of fetal fibronectin (FFN) testing and serial cervical length (CL) surveillance conducted every 1–2 weeks. The synergistic application of these tests enhances predictive accuracy beyond what each test can achieve individually. Specifically, a negative FFN result coupled with a normal cervical length may justify the extension of outpatient expectant management until the 36–37-week mark. This not only fosters reassurance for both patient and physician but also holds the potential for healthcare cost savings. By integrating FFN and CL surveillance into vasa previa management protocols, healthcare providers can proactively assess risks, intervene early and tailor care strategies to individual patient needs. This approach ultimately aims to optimize outcomes for both mother and fetus in instances of this rare yet serious obstetric complication.

## 8. Conclusions and Future Considerations

Vasa previa is a pregnancy complication in which unprotected fetal blood vessels traverse the cervical os, placing the fetus at high risk of exsanguination and fetal death. The fetal vessels may be compromised by fetal movement and compression, leading to poor oxygen distribution and asphyxiation. FFN provides insight on the maternal and fetal membrane connection stability, while serial CL surveillance can offer information on the increased risk of cervical shortening, ultimately leading to preterm labor [30]. There is currently limited evidence of FFN and CL surveillance in the prediction and management of preterm delivery in the setting of vasa previa. The studies utilizing FFN and CL surveillance have shown an improvement in the prediction of preterm labor, especially when the CL measurement was within the borderline range or there were symptoms of preterm labor [3]. However, in other studies, the combination of FFN and CL surveillance showed a neutral impact on the prediction of preterm labor [36]. One can consider extrapolating these findings as there may be some benefit for vasa previa patients to undergo management with FFN and CL surveillance every 1–2 weeks from 24 weeks’ to 36 weeks’ gestation until delivery. Due to the high danger associated with PTL and vasa previa, and the possible consequences associated with preterm delivery, strong considerations and future studies should assess the utilization and impact of FFN and CL surveillance in the management of vasa previa. Moreover, exploring the potential integration of other biomarkers or imaging modalities alongside FFN and CL surveillance could enhance the predictive accuracy and precision of vasa previa management strategies. Both FFN and CL surveillance are generally considered low-risk in terms of their direct impact on pregnancy. However, it is crucial to recognize the emotional toll this surveillance can take on expectant mothers awaiting their results. The anticipation of such pivotal information can evoke heightened emotions, leading to elevated levels of stress and anxiety among patients. Thus, while the physical risks may be minimal, the psychological impact of FFN testing should not be overlooked, necessitating empathetic support and guidance for patients throughout the process. Ultimately, a comprehensive understanding of the role of FFN and CL surveillance in vasa previa management is crucial for optimizing prenatal care and minimizing adverse outcomes associated with this rare but high-risk obstetric condition.

Future considerations for vasa previa encompass a multidimensional approach encompassing advancements in diagnostic modalities, refinement of management protocols, development of risk prediction models, evaluation of conservative management strategies, implementation of multidisciplinary care models, patient education and long-term outcome studies. By addressing these considerations, healthcare providers can strive towards optimizing the care and outcomes of pregnancies complicated by vasa previa.

## Figures and Tables

**Figure 1 diagnostics-14-01016-f001:**
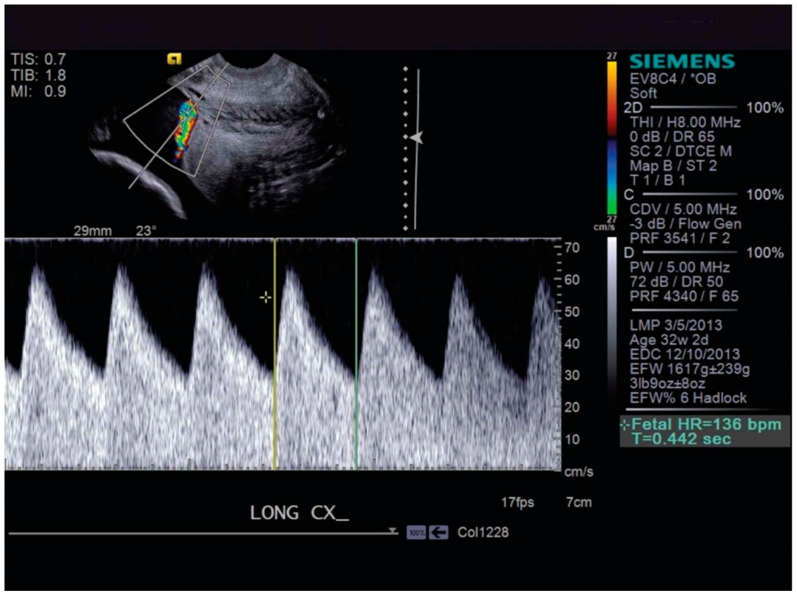
Color Doppler demonstrating fetal vessels over the cervix. Silver, Placenta Previa, Vasa Previa and Placenta Accreta. Obstet. Gynecol. 2015 [13].

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
