# Peer review of "Vasa Previa and the Role of Fetal Fibronectin and Cervical Length Surveillance: A Review"

_diagnostics, 2024, doi:10.3390/diagnostics14101016_

Round 1

Reviewer 1 Report

Comments and Suggestions for Authors

The authors reported reviews in management of vasa previa using FFN and CL. They recommended FFN testing and serial CL surveillance every 1-2 weeks. Manuscript is well written and satisfied to review. Besides, I completely agree with CL measurement, but in my opinion I do not recommend to use FFN for management of vasa previa. False positive of FFN might result in artificial preterm delivery. Otherwise, we do not use FFN even for predicting preterm delivery itself. Recent reports of vasa previa suggested that increase of perinatal mortality in antenatal diagnosed cases was not increased, outpatient management and CS at 36 weeks' gestation was allowable in asymptomatic women.

Author Response

I would like to express my sincere gratitude for your thoughtful and constructive feedback on Vasa Previa and the Role of Fetal Fibronectin and Cervical Length Surveillance: A Review. We appreciate the time and effort you dedicated to the thorough review. In response to your suggestions, we have carefully revisited the manuscript and made necessary revisions to address the concerns raised. In this letter, we provide a detailed account of the changes implemented, explaining how each modification contributes to enhancing the clarity, coherence, and robustness of our work. Thank you once again for your invaluable input; we look forward to your continued guidance as we strive to make this contribution to the literature even more impactful.

Thank you so much for the feedback on the paper. We agree that inaccurate FFN results and occur and cause distress to patients. We are suggesting to consider it in the management of vasa previa. We expanded on this further in regards of the false positive and how that may effect further management.

Reviewer 2 Report

Comments and Suggestions for Authors

This is an interesting study. However, irrespectively of the title of the manuscript, in terms of “management”,  some more actions should be included. Examples: 1. (at least) daily nonstress test at 30-34 weeks of gestation after patient’s admission to the hospital. 2. A course of antenatal corticosteroids by 34 weeks. 

Author Response

I would like to express my sincere gratitude for your thoughtful and constructive feedback on Vasa Previa and the Role of Fetal Fibronectin and Cervical Length Surveillance: A Review. We appreciate the time and effort you dedicated to the thorough review. In response to your suggestions, we have carefully revisited the manuscript and made necessary revisions to address the concerns raised. In this letter, we provide a detailed account of the changes implemented, explaining how each modification contributes to enhancing the clarity, coherence, and robustness of our work. Thank you once again for your invaluable input; we look forward to your continued guidance as we strive to make this contribution to the literature even more impactful.

We  expanded more on the management options including antenatal steroids and NST. 

Round 2

Reviewer 1 Report

Comments and Suggestions for Authors

I think this is acceptable for publication in current form.

Author Response

I want to express my deep appreciation for the thoughtful and helpful feedback you provided on our review of Vasa Previa and the Role of Fetal Fibronectin and Cervical Length Surveillance. Your insights were incredibly valuable, and we are grateful for the time and effort you put into reviewing our work.

Based on your suggestions, we have carefully gone through the manuscript and made the necessary revisions to address the issues you raised.Thank you once again for your invaluable input. We eagerly anticipate your ongoing guidance as we continue to refine our contribution to the literature and strive to make it even more impactful.